

# Psychometric evaluation of the Chinese version of Risky Loot Box Index (RLI) and cross-sectional investigation among gamers of China

Peidong Guo, Yueheng Liu, Luyin Tan, Yifan Xu, Haolin Huang and Qijian Deng

Department of Psychiatry, National Clinical Research Center for Mental Disorders, and National Center for Mental Disorders, The Second Xiangya Hospital of Central South University, Hunan, China

## ABSTRACT

Nowadays, many of the top-selling video games include options to purchase loot boxes as paid virtual items. As research progressed, loot boxes have been found to have similar characteristics to gambling, and there has been an ongoing debate as to whether loot boxes can be defined as gambling. In order to better study loot boxes, psychometrically meaningful scales are necessary. The Risky Loot Box Index (RLI) was developed by Brooks and Clark, which is the most commonly used tool to assess the use of loot boxes. This study aimed to translate the original RLI into Chinese and evaluate its psychometric properties. Two samples were recruited through online gaming forums ($n$ = 143) and offline internet cafes ($n$ = 236). An exploratory factor analysis of the online sample yielded a one-dimensional nine-item model, with the factor focused on risky behaviors associated with loot boxes. The confirmatory factor analysis carried out on the offline sample corroborated the results obtained from the exploratory factor analysis, and the Chinese version of the RLI displays satisfactory psychometric properties. Furthermore, the Problem Gambling Severity Index (r = 0.57, $P < 0.001$) and the Internet Gaming Disorder Scale-Short Form (r = 0.67, $P < 0.001$) were found to be significantly associated with the RLI. We also found that players with high RLI scores may have higher levels of anxiety and depression, and they were more willing to spend money on loot boxes, with some spending nearly all their earnings. Interestingly, no significant correlations between age, gender, education, or income level, and the RLI were found.

# INTRODUCTION

With the popularity of the Internet, various activities derived from it, such as online videos, shopping, social media, and gaming, have become part of people's lives. Among these, video games have become a major form of entertainment for people worldwide. The Newzoo Global Games Market Report 2023 states that the number of global gamers has reached 3.38 billion in 2023 (*Newzoo, 2024*). Large user groups have given rise to an extensive gaming market. In the first half of 2024, Chinese domestic game market revenue

Corresponding author
Qijian Deng, dengqijian@csu.edu.cn

figures hit another record high of 147.267 billion yuan, while the game user scale of 674 million was also the highest in history (*China Game Industry Association (CGIGC), 2024*).

Over the past decade, a new profit model has emerged in addition to direct game sales. Instead of selling all game content at once, game publishers offer core game content to attract users and then progressively add various optional paid content to the game to generate revenue (*Hamari et al., 2017*). In gaming terminology, this form of in-game purchase is referred to as a 'micro-transaction' (*Tomic, 2017*). Micro-transactions include a variety of content, such as clothing or skins, battle passes, and loot boxes (*Gibson et al., 2023*), the latter being the main type of micro-transactions which can be found in many top-selling video games. More than half of the top games on Google Play and Appstore, and more than a third of these on Steam contain loot boxes (*Zendle et al., 2020*). In mainland China, 91 of the 100 highest-grossing Chinese iPhone games contain loot boxes (*Xiao et al., 2024b*).

Usually, loot boxes can be purchased with in-game currency or real money, and players can open them to obtain virtual items (*e.g.*, in-game props, skins, characters). Moreover, the virtual items are random and the players do not know which rewards they will obtain until the loot boxes are opened. Due to their 'monetary' and 'random' nature, there is an ongoing debate as to whether loot boxes are a form of gambling. Drummond and Sauer conducted a systematic analysis of 22 video games containing loot boxes to determine whether they fit gambling characteristics. The results showed that nearly half of them fully met all gambling characteristics, and the vast majority met most of them (*Drummond & Sauer, 2018*). However, not all games can sell virtual goods (or profit from loot boxes), making it difficult to define whether using loot boxes is gambling.

Although the debate on whether loot boxes are a gambling form is ongoing, their negative effects have been proven. A considerable body of research indicates a positive correlation between loot box use and problem gambling. On one hand, loot boxes may be a trigger for problem gambling; on the other hand, individuals with a predisposition to problem gambling are more likely to engage in loot box activity (*Close et al., 2022*; *Kristiansen & Severin, 2020*; *Zendle & Cairns, 2019*). In addition, players who purchased loot boxes played video games longer and more frequently. They also had a higher severity of problem gaming and were more likely to meet the diagnostic criteria for Internet Gaming Disorder (IGD; *Garea et al., 2021*; *Li, Mills & Nower, 2019*). Moreover, buying loot boxes was associated with higher levels of mental stress (*Li, Mills & Nower, 2019*) as well as negative emotions such as depression, anxiety, and guilt (*Sanmartín et al., 2023*; *Shinkawa et al., 2021*), and showed more behavioural and peer problems in terms of psychosocial adjustment (*Shinkawa et al., 2021*).

So far, most studies have assessed loot box usage through a single metric: 'money spent on loot box purchases' (*Yokomitsu et al., 2021*). However, simply assessing loot box purchases ignores other relevant risky behaviours and negative consequences (*Forsström et al., 2022*). To address this problem, Brooks and Clark developed a 12-item scale, the Risky Loot Box Index (RLI), in which higher scores represent a higher risk of loot box use (*Brooks & Clark, 2019*). The RLI has been widely used in several countries, including

Canada (*Brooks & Clark, 2019*), Sweden (*Forsström et al., 2022*) and Poland (*Cudo, Montag & Pontes, 2024*), and has demonstrated good reliability and validity in comprehensively assessing loot box usage.

In 2023, *Xiao et al. (2024a)* translated the RLI into Chinese for the first time, basing on the original version (one-dimension, five-item) developed by *Brooks & Clark (2019)*. They then conducted a confirmatory factor analysis (CFA), but the results were not entirely satisfactory and revealed that the Chinese version of the RLI could be improved. Additionally, *Forsström et al. (2022)* proposed a new Swedish version of the RLI (two-dimensional, seven-item) after re-conducting an exploratory factor analysis (EFA) on the original 12-item scale. This new version differs from the original single-dimensional scale. Given these issues, which may be due to cultural variations or other factors across different countries, it is necessary to retranslate the original 12-item scale into Chinese and conduct an EFA to ensure its validity and reliability in the Chinese context.

Therefore, we aimed to translate the RLI into Chinese. Then, the factor structure, validity, and reliability of the RLI-C were evaluated among Chinese video gamers, and the impact factors related to loot box usage were explored.

## MATERIALS AND METHODS

### Procedure

The study data were collected through online and offline surveys using the same questionnaire, which was created on the 'Questionnaire Star' website. The online survey was posted on a popular Chinese online forum called 'Baidu Tieba', specifically targeting two of the most popular video game sub-forums ('Genshin Impact' and 'Counter-Strike'). We created new recruitment posts in subforums visible to other users. The questionnaire link was embedded directly into the posts and participants click on them to access the survey. We then monitored the posts regularly, and periodically reposted recruitment messages to increase visibility.

Offline, players in internet cafés in Changsha (a city in China) were invited to complete the questionnaire. In response to the observation that most gamers in internet cafés are reluctant to participate when approached directly because of their immersion in gaming activities, we developed a non-intrusive strategy, allowing participation at their convenience. We designed small cards containing a brief introduction to our research and a QR code (scanning linked to an online questionnaire). Interested individuals could take a card and complete the survey at a convenient time. This strategy significantly minimised the disruption to the players' gaming experience and increased their participation rates.

All participants who completed the questionnaire were rewarded ¥1 (approximately $0.15) by the questionnaire system after the review and approval of the research team.

When participants opened the questionnaire, they were informed about the study's background, purpose, as well as its voluntary and anonymous nature, and asked to provide their consent. The questionnaire took approximately 5 min to complete. It included demographic questions (age, sex, and education), video game and loot box-related questions, and the following six scales, which are detailed in the *Measures* section: RLI

(*Brooks & Clark, 2019*), IGDS-9SF (*Leung et al., 2020*; *Yam et al., 2019*), PGSI (*Loo, Oei & Raylu, 2011*), BIS-Brief (*Luo et al., 2020*), PHQ-9 (*Wang et al., 2014*), GAD-7 (*He et al., 2010*). This study was approved by the ethics committee of the Second Xiangya Hospital Clinical Research Center (LYF20240168). The surveys were conducted for three months, between 1 June and 1 September 2024.

For quality control, the following criteria were used to remove invalid questionnaires: (1) answer time < 120 s; (2) wrong answer to 'trap' question, such as 'the result of 2 + 3'; (3) the same option for ten consecutive questions. In addition, questionnaires were excluded if respondents indicated (1) they I had not played any games recently (was not a gamer), or (2) they were under 18 years of age.

## Participants

A total of 527 questionnaires were collected. After applying our exclusion criteria, 379 valid samples were included in the analysis. Specifically, for the online survey, a total of 109 invalid questionnaires were excluded (99 completed too quickly, one had the wrong answer to trap questions, two were not gamers, four provided unreasonable answers such as 'I play games for 500 h per week', and three were identified as potential robot responses due to identical IP addresses, response times, and answers). From the offline surveys, 39 invalid questionnaires were excluded (33 completed too quickly, two had wrong answers to trap questions, two were not gamers, and two provided unreasonable answers).

Of the final 379 valid samples included in the analysis, 143 were from the online survey and 236 were from the offline survey. Notably, some excluded questionnaires met multiple exclusion criteria. Therefore, we applied the criteria sequentially, removing each questionnaire based on the first applicable criterion. Table 1 shows the detailed demographic information of participants.

## Measures

### Video game & loot box-related questions

For the video game-related questions, we used a more general question, 'Have you played video games recently?', to determine whether the participants were gamers, and we asked them the number of hours they played per week. For the loot box-related questions, we collected the participants' monthly monetary expenditure on loot boxes and their monthly income. Detailed information regarding these questions is presented in Table 2.

### Chinese version of Risky Loot Box Index (RLI-C)

The RLI is widely used to evaluate the risk of gamers' loot box purchases and usage. The original scale was made by *Brooks & Clark (2019)* and has 12 items, rated on a five-point Likert scale. The total score ranges from 12 to 60, with higher scores indicating a higher risk of loot box purchases and usage. After obtaining permission from the original authors, the RLI was translated based on the Brislin translation model (*Brislin, 1970*). First, two native Chinese-bilingual psychologists independently translated the items from English into Chinese. A discussion group was convened to discuss and revise all items, resulting in a consistent forward Chinese version. Second, the forward Chinese version was

**Table 1 Demographic data.**

| Demographics | Sample 1: (Online survey) *n* = 143 | Sample 2: (Offline survey) *n* = 236 |
|---|---|---|
| Age: | | |
| Mean (SD) | 25.36 (4.63) | 25.25 (4.49) |
| Gender (%): | | |
| Male | 109 (76.2%) | 172 (72.9%) |
| Female | 34 (23.8%) | 64 (27.1%) |
| Education (%): | | |
| High School or lower | 7 (4.9%) | 11 (4.7%) |
| Junior college | 21 (14.7%) | 49 (20.8%) |
| Undergraduate | 87 (60.8%) | 170 (72.0%) |
| Postgraduate | 28 (19.6%) | 6 (2.5%) |

**Table 2 Video game & loot box-related data.**

| Question | Sample 1: (Online survey) *n* = 143 | Sample 2: (Offline survey) *n* = 236 |
|---|---|---|
| Weekly gaming time (hours) | 17.8 (14.6) | 20.7 (15.2) |
| Monthly monetary expenditure on loot boxes (CNY, ¥) | 606.6 (1,946.9) | 623.1 (744.5) |
| Monthly income (CNY, ¥) | 6,619.0 (10,148.7) | 5,743.0 (4,689.3) |

independently back translated into English by two native Chinese scholars who did not know the RLI before. Third, the back-translated version was emailed to the original author for advice and appropriate revisions were made to ensure consistency between the original and back-translated versions. Finally, an item review was conducted by an expert panel of 12 psychologists, and revisions were made based on the item review. The RLI-C showed good internal consistency in both the online ($\alpha$ = 0.93) and offline samples ($\alpha$ = 0.86).

### Internet Gaming Disorder Scale-Short Form (IGDS-9SF)

The Chinese version of the IGDS-9SF is a self-rating scale used to assess the severity of IGD symptoms (*Leung et al., 2020*; *Yam et al., 2019*). The items of the IGDS-9SF are rated on a five-point Likert scale ranging from 1 (*never*) to 5 (*very often*). The scores range from 9 to 45, with a higher score indicating a higher degree of disordered gaming. The IGDS-9SF demonstrated good internal consistency both in the online ($\alpha$ = 0.93) and offline samples ($\alpha$ = 0.88) in the IGDS-9SF.

### Problem Gambling Severity Index (PGSI)

Gambling is strictly prohibited in mainland China under criminal law. This prohibition extends to all forms of gambling that are not explicitly sanctioned or organised by the government. It is pertinent to highlight that only lottery activities authorised and managed by governmental bodies are considered legal under Chinese law (*National People's Congress, 2020*). In 2005, the Supreme People's Court and Supreme People's Procuratorate clarified that non-profit-oriented betting activities involving small stakes were not

considered gambling crimes (*Wu & Lau, 2015*). Therefore, people in mainland China are more inclined to regard gambling with small stakes as a recreational 'game' than simply gambling.

In this context, the Chinese people's understanding of the term 'gambling' ('赌博') differs from the international norm (*Keovisai & Kim, 2019*). To address this, we referred to Xiao's research design (*Xiao, Fraser & Newall, 2022*). The definition of 'gambling' was introduced briefly to participants in the questionnaire (Fig. 1) so that they could have an accurate understanding of gambling activities rather than seeing it as a recreational game. The PGSI, the world's most widely used self-scoring tool, was used to assess problem gambling. Higher scores indicate worse gambling disorders. *Loo, Oei & Raylu (2011)* translated the original version of the PGSI into traditional Chinese. After obtaining the author's permission, we translated this scale from traditional Chinese into Simplified Chinese and made minor changes in grammar and vocabulary to adapt to the language habits of mainland China. In this study, the consistency of the PGSI was excellent in both samples (online, $\alpha = 0.96$; offline, $\alpha = 0.93$).

### Barratt Impulsiveness Scale-Brief (BIS-Brief)

The BIS-Brief was developed by *Steinberg et al. (2013)*. The scale was translated into Chinese by *Luo et al. (2020)* and the copyright holders granted permission to use this instrument. The scale consists of eight items from the BIS-11 and each item is scored on a four-point Likert scale. The total BIS-Brief scores range from 8 to 32, with higher scores indicating higher impulsivity. The scale showed good internal consistency both in the online ($\alpha = 0.81$) and offline samples ($\alpha = 0.86$).

### Nine-item Patient Health Questionnaire (PHQ-9)

*Spitzer, Kroenke & Williams (1999)* designed the PHQ-9 with reference to the Diagnostic and Statistical Manual of Mental Disorders (DSM-IV) diagnostic criteria for major depressive disorder. The PHQ-9 uses a four-point Likert scale, with total scores ranging from 0 to 27. A higher score indicating a higher level of depression. The scale showed good internal consistency in the online ($\alpha = 0.89$) and offline samples ($\alpha = 0.93$).

### Seven-item Generalized Anxiety Disorder Scale (GAD-7)

The GAD-7 was designed by *Spitzer et al. (2006)* and was derived from the DSM-IV diagnostic criteria for generalised anxiety disorder. This seven-item scale, uses a four-point Likert scale with total scores ranging from 0 to 21. A higher score indicates a higher severity of anxiety. The scale demonstrated excellent internal consistency in both the online ($\alpha = 0.93$) and offline samples ($\alpha = 0.93$).

## Statistical analysis

Qualitative data were presented as percentage (%) and quantitative data were presented as mean ± standard deviation (SD). SPSS (version 24.0) and AMOS (version 24.0) were used for statistical analysis. $P < 0.05$ was considered statistically significant.

An item analysis was conducted to ensure the effectiveness of the items. The structural validity of the RLI-C was tested *via* EFA and CFA. EFA was conducted using an online

请根据你过去 12 个月中的赌博经历来回答下述问题

这里的赌博指：

1. 购买彩票（足彩、体彩、福彩）或者刮刮乐
2. 任何涉及金钱的赌注形式（包括但不局限于与亲朋好友打麻将、扑克牌、电子竞猜等）

The following questions refer to your gambling in the past 12 months.

Gambling here means:

1. Purchase lottery tickets (football lottery, sports lottery, charitable lottery) or scratch cards.
2. Any form of betting involving money (including, but not limited to, friendly Mahjong, poker, e-sport betting).

**Figure 1 The Chinese and English version of the definition introduction of 'gambling' in the questionnaire.**

sample and principal axis factoring was employed because of the presence of skewness in the data. CFA was conducted using the AMOS software based on the offline sample. Content validity was assessed using the Delphi method.

For the test-retest reliability analysis, an additional question was added at the end of the questionnaire: 'Would you be willing to participate in a follow-up survey?' We collected the phone numbers of participants who answered 'yes'. From the pool of online survey participants who responded affirmatively, we randomly selected 50 individuals to complete the scale again 1 month later, and a test-retest reliability analysis was completed based on the results.

## RESULTS

### Item analysis

The effectiveness of each item was tested using the online sample. First, the RLI scale was divided into high and low groups based on total scores, with the first 27% classified as high and the last 27% as low. The two groups were analysed using an independent sample $t$-test. Items that demonstrated no statistically significant differences ($P > 0.05$) were excluded. The correlation coefficients between each item and the overall score were tested using the Pearson correlation analysis. Items with coefficients > 0.4 showed a satisfactory correlation with the scale.

As a result, each item in RLI-C showed a statistical difference (Table 3). The item-total score correlation was found with each item. All of them had good correlation coefficients, which ranged from 0.55–0.84 (Table 4).

### Exploratory factor analysis

The exploratory factor analysis (EFA) analyses were based on an online sample ($n = 143$), following Thompson's recommendation of 10–20 people per measure (*Thompson, 2004*). Before the analysis, Bartlett's test of sphericity and the Kaiser-Meyer-Olkin (KMO) test were performed. Bartlett's test of sphericity was 1,159 (df = 66, $P < 0.001$) and the KMO index was 0.92. Thus, the data were considered suitable for the subsequent factor analysis.

**Table 3 Score comparison between high-score and low-score groups of the Chinese version of RLI.**

| Item | Low-score group (n = 40) | | High-score group (n = 43) | | CR | P |
|------|------|------|------|------|------|------|
| | Mean | SD | Mean | SD | | |
| Q1 | 1.88 | 0.91 | 4.37 | 0.62 | 14.50 | <0.001 |
| Q2 | 1.80 | 1.04 | 4.00 | 0.85 | 10.51 | <0.001 |
| Q3 | 1.75 | 0.84 | 4.16 | 0.72 | 14.07 | <0.001 |
| Q4 | 1.70 | 0.82 | 4.12 | 0.73 | 14.17 | <0.001 |
| Q5 | 1.85 | 1.12 | 4.26 | 0.82 | 11.09 | <0.001 |
| Q6 | 1.93 | 1.25 | 3.65 | 1.23 | 6.34 | <0.001 |
| Q7 | 1.30 | 0.52 | 3.93 | 0.91 | 16.03 | <0.001 |
| Q8 | 1.78 | 1.03 | 4.14 | 0.64 | 12.50 | <0.001 |
| Q9 | 1.30 | 0.76 | 3.98 | 1.26 | 11.80 | <0.001 |
| Q10 | 1.28 | 0.55 | 3.98 | 0.77 | 18.21 | <0.001 |
| Q11 | 1.33 | 0.66 | 3.95 | 0.90 | 15.13 | <0.001 |
| Q12 | 1.33 | 0.76 | 3.47 | 1.22 | 9.64 | <0.001 |

**Table 4 Item-total score correlation of the Chinese version of RLI (n = 143).**

| Item | r | P |
|------|------|------|
| Q1 | 0.80 | <0.001 |
| Q2 | 0.73 | <0.001 |
| Q3 | 0.84 | <0.001 |
| Q4 | 0.79 | <0.001 |
| Q5 | 0.74 | <0.001 |
| Q6 | 0.55 | <0.001 |
| Q7 | 0.81 | <0.001 |
| Q8 | 0.73 | <0.001 |
| Q9 | 0.77 | <0.001 |
| Q10 | 0.84 | <0.001 |
| Q11 | 0.80 | <0.001 |
| Q12 | 0.71 | <0.001 |

As the data were skewed, principal axis factoring was used for the analysis (*Costello & Osborne, 2005*). Following the suggestions of *Thompson (2004)*, one item was excluded because of high collinearity (>0.8) and two were excluded because of low communality (<0.45). This procedure was repeated after removing each item. The remaining nine items were suitable for factor analysis (KMO = 0.91; Bartlett's test of sphericity, $X^2$ = 853, df = 36, $P < 0.001$). Factors with eigenvalues > 1 were retained, and a single-factor solution was produced. The square loadings were 5.3 and explained 59.3% of total variance. This was confirmed using a scree plot (Fig. 2). Details of the factor loadings and communalities for

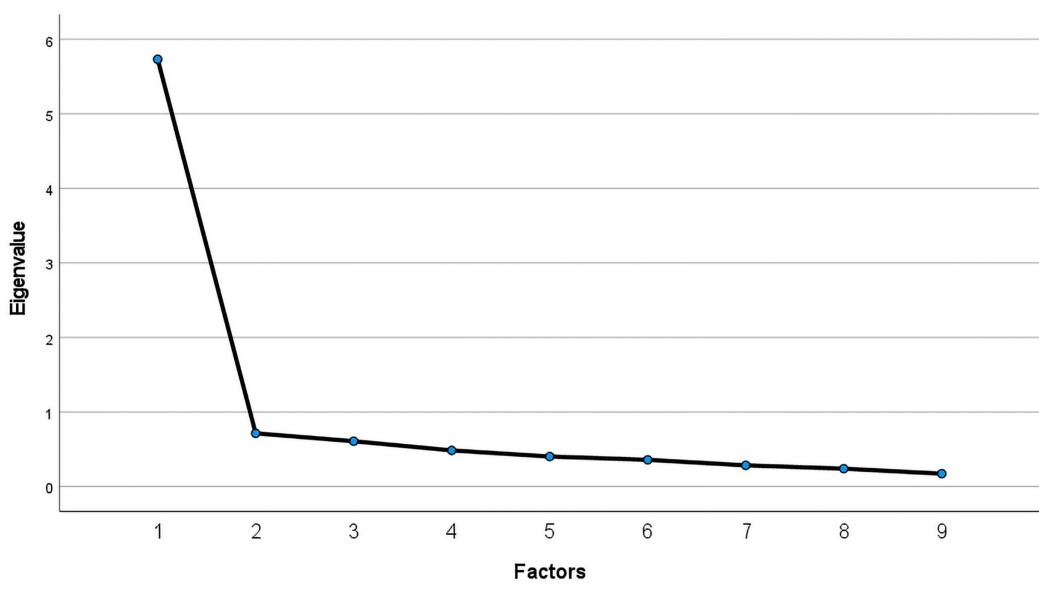

**Figure 2  Scree plot of Chinese version of RLI.**   

**Table 5  Factor loadings and communalities for each item of Chinese version of RLI.**

| Items | Factor loading | Communality |
|---|---|---|
| 1. I frequently play games longer than I intend to, so I can earn Loot Boxes. | 0.80 | 0.64 |
| 2. I believe obtaining items from Loot Boxes is an effective way to generate money. | 0.74 | 0.55 |
| 3. I will play for long periods of time to earn Loot Boxes. | 0.86 | 0.74 |
| 4. Receiving items from Loot Boxes is a primary reason why I play video games. | 0.79 | 0.63 |
| 5. I buy Loot Boxes with the hope of receiving valuable items to sell. | 0.72 | 0.51 |
| 6. I have put off other activities, work, or chores to be able to earn or buy more Loot Boxes. | 0.80 | 0.63 |
| 7. Once I open a Loot Box, I often feel compelled to open another. | 0.69 | 0.48 |
| 8. I have sometimes spent more on Loot Boxes than I could afford. | 0.71 | 0.50 |
| 9. I have bought more Loot Boxes after failing to receive valuable items. | 0.81 | 0.65 |

each retained item are listed in Table 5. Cronbach's α indicated excellent internal consistency (α = 0.93). Similar to the original authors (*Brooks & Clark, 2019*), this variable was primarily used to assess the risky behaviours of loot boxes.

## Confirmatory factor analysis

A CFA was employed on the offline sample ($n$ = 236) to corroborate the model derived from the EFA. These results (Table 6) indicated a good fit for the model, following *Thompson*'s *(2004)* recommendations. A graphical representation of this is shown in Fig. 3. The factor loadings for each item ranged from 0.46 to 0.77 ($P < 0.05$), indicating a statistically significant correlation.

**Table 6 Goodness-of-fit indices of the one-factor model.**

| $\chi 2$ /df | RMSEA | NFI | IFI | TLI | CFI | GFI | RMR |
|---|---|---|---|---|---|---|---|
| 2.503 | 0.080 | 0.907 | 0.942 | 0.922 | 0.941 | 0.938 | 0.059 |

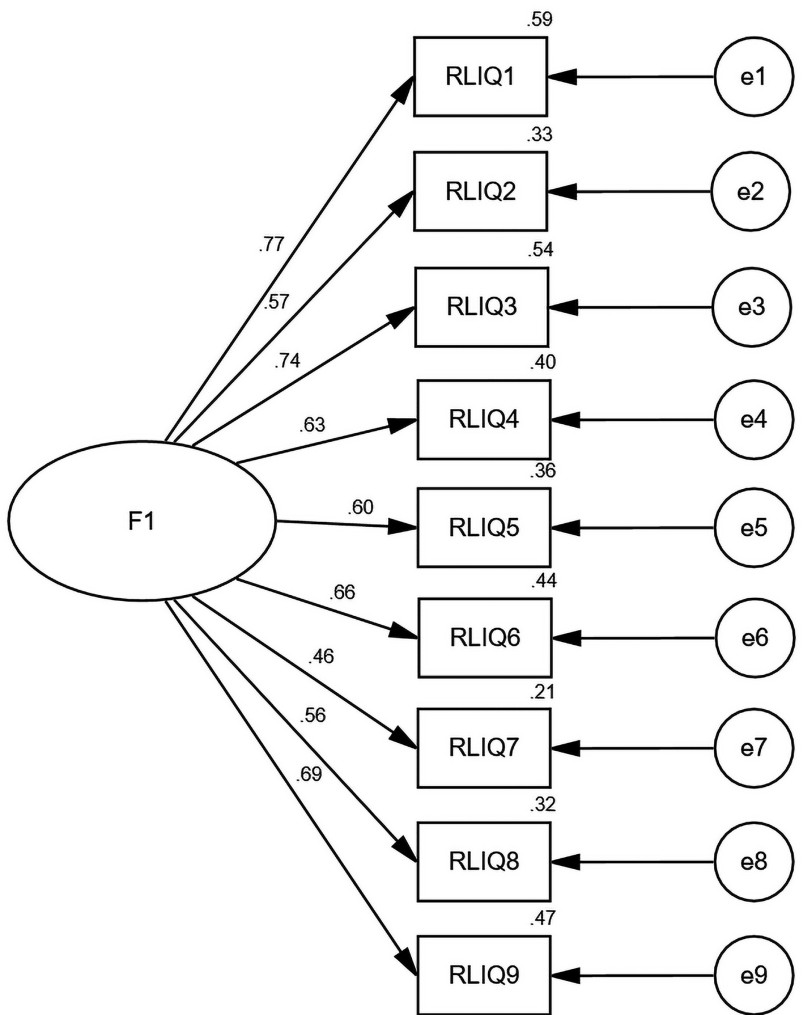

**Figure 3 CFA results of the Chinese version of RLI.** F1 represents risky behaviors of loot boxes.

## Content validity

The scale was evaluated by an expert panel of six psychiatrists. The Item Content Validity Index ranged from 0.83 to 1.00, and the average scale content validity index (S-CVI/ Ave) was 0.94. Both results demonstrated satisfactory content validity.

## Criterion-related validity

The PGSI and IGDS-9SF were used to assess criterion-related validity. Spearman's correlation analysis was used because of the presence of skewed data. The results showed

**Table 7 Criterion-related validity of the Chinese version of RLI (n = 379).**

| | Revised RLI | PGSI | PGSI (Problem gambling behaviors) | PGSI (Negative consequences) | IGDS-9SF |
|---|---|---|---|---|---|
| Revised RLI | – | – | – | – | – |
| PGSI | 0.57** | – | – | – | – |
| PGSI (Problem gambling behaviors) | 0.57** | 0.95** | – | – | – |
| PGSI (Negative consequences) | 0.52** | 0.96** | 0.83** | – | – |
| IGDS-9SF | 0.67** | 0.65** | 0.65** | 0.60** | – |

**Note:**
** Significant correlation at 0.01.

that the RLI-C positively correlated with the total score of the PGSI, two-dimensional scores of the PGSI, and the total score of the IGDS-9SF. The results are presented in detail in Table 7.

### Reliability

For Cronbach's α coefficient, the RLI-C showed excellent consistency both in online (α = 0.93) and offline samples (α = 0.86). The overall test-retest reliability of the RLI-C was 0.78 ($P < 0.05$).

### Correlation analysis

In addition to being associated with the IGSF and PGSI, the Chinese version of the RLI was associated with weekly game time (0.20, $P < 0.01$), monthly loot box spending (0.47, $P < 0.01$), BIS-Brief (0.14, $P < 0.01$), GAD-7 (0.45, $P < 0.01$), and PHQ-9 (0.43, $P < 0.01$).

### DISCUSSION

Similar to *Brooks & Clark (2019)*, our EFA results replicated the one-factor model. In total, three items were deleted: Item 11 'The thrill of opening Loot Boxes has encouraged me to buy more' was deleted due to high collinearity (>0.8); Item 12 'My loot box use has caused me problems' and Item 6 'I have felt guilty about the amount of time or money I have spent on Loot Boxes' were deleted due to low communality (<0.45).

Interestingly, except for Item 11, all deleted items were associated with negative effects. The remaining nine items were associated with risky behaviours that may become problematic. Consistent with the original author's opinion, we defined this dimension as the risky behaviour of loot boxes.

More items were retained compared to the original scale created by Brooks and Clark. The main reason for this discrepancy may be the difference in the samples. The samples for this study came directly from video game forums and internet cafés. More than 95% of the respondents had gaming experience, which was higher than in the original study (84.8%). Stricter screening criteria were used to ensure sample quality. Thus, our sample was more suitable for investigating loot box usage. Cultural differences may also be the reason. 'Loot boxes' are not widely used in China yet; this makes the term 'loot boxes' difficult to understand when translated into Chinese. *Xiao et al. (2024a)* also noted this issue. To address this, the questionnaire provided a thorough explanation of loot boxes, including

images from popular online games. However, some participants may still have misunderstood the concept. This potential cognitive bias may have affected the results of the final scale. Overall, the CFA results showed that the fit indices of the one-factor model met the statistical requirements and were valid across different populations. Thus, the RLI-C had excellent construct validity.

The RLI was positively correlated with the IGDS-9SF and PGSI, suggesting that the loot box, as a fused form of gambling and gaming, is strongly associated with both. Similar to individuals with gambling or gaming disorders, those with high loot box usage experience psychological stress. Furthermore, a low positive correlation was found between the RLI and BIS-Brief scores, suggesting that risky loot box usage may be associated with poor impulse control.

In addition, this study explored the relationship between the RLI and sex, age, and education level. Surprisingly, none of these correlated with the RLI. This conclusion is consistent with previous research (*Drummond, Hall & Sauer, 2022*). This means that, although youngers, boys are more involved in video games and loot boxes, this does not mean that they are more likely to have risky use of loot boxes.

Another interesting aspect that we found was that the RLI was not associated with monthly income, which is consistent with the result of a previous study (*Close et al., 2021*). Instead, the RLI was strongly associated with monthly monetary expenditure on loot boxes. Moreover, the higher the RLI score, the higher was the proportion of monthly loot boxes spent on monthly income. Some players spent almost all of their income on loot boxes.

### Limitations and further research

First, owing to the strict ban on gambling behaviour in mainland China (both online casinos and offline gambling are illegal), it was difficult to collect a credible and sufficient sample of gamblers for this study. Further studies should consider sample collection in regions of China where gambling is not prohibited (*e.g.*, Macau) to further validate the revised RLI.

Second, our study only included participants over the age of 18 years, whereas a large percentage of adolescents are exposed to loot boxes. According to the article by *Montiel et al. (2022)*, the prevalence of loot box purchases for adolescent gamers is 20–33.9%. Therefore, it is crucial to test the validity and reliability of the RLI in adolescents.

## CONCLUSIONS

This study provides an effective instrument for assessing risky loot box behaviours within the Chinese cultural context and background, laying a solid foundation for addressing this issue. The Chinese version of the RLI shows excellent reliability and validity and can be utilised in the future for the preliminary screening of high-risk populations, providing a reliable theoretical basis for the development and execution of subsequent targeted intervention plans.

## ACKNOWLEDGEMENTS

We would like to express our gratitude to Professor Gabriel A. Brooks from Centre for Gambling Research at UBC, Department of Psychology, University of British Columbia, 2136 West Mall, Vancouver, BC V6T 1Z4, Canada. We thank him for providing the RLI and agreeing to revise it, and for making important comments during the revision process.

### Funding

This work was supported by the Science and Technology Major Special Fund Project of Changsha (No. kh2401006). The funders had no role in study design, data collection and analysis, decision to publish, or preparation of the manuscript.

### Grant Disclosures

The following grant information was disclosed by the authors:
Science and Technology Major Special Fund Project of Changsha: kh2401006.

### Competing Interests

The authors declare that they have no competing interests.

### Author Contributions

- Peidong Guo conceived and designed the experiments, performed the experiments, analyzed the data, prepared figures and/or tables, authored or reviewed drafts of the article, and approved the final draft.
- Yueheng Liu performed the experiments, authored or reviewed drafts of the article, and approved the final draft.
- Luyin Tan performed the experiments, prepared figures and/or tables, and approved the final draft.
- Yifan Xu performed the experiments, prepared figures and/or tables, and approved the final draft.
- Haolin Huang performed the experiments, prepared figures and/or tables, and approved the final draft.
- Qijian Deng conceived and designed the experiments, analyzed the data, authored or reviewed drafts of the article, and approved the final draft.

### Human Ethics

The following information was supplied relating to ethical approvals (*i.e.*, approving body and any reference numbers):

The Second Xiangya Hospital Clinical Research Center Ethics Committee (LYF20240168).

### Data Availability

The raw data is available in the Supplemental Files.

## Supplemental Information

Supplemental information for this article can be found online at http://dx.doi.org/10.7717/peerj.19164#supplemental-information.

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
