# Peer review of "Psychometric evaluation of the Chinese version of Risky Loot Box Index (RLI) and cross-sectional investigation among gamers of China"

_PeerJ, doi:10.7717/peerj.19164_

## Round 0.1 · original submission · Major Revisions

I have completed my evaluation of your manuscript. The reviewers recommend reconsideration of your manuscript following major revision. I invite you to resubmit your manuscript after addressing the comments below

Reviewer 1 ·

Basic reporting

The very first sentence of the abstract (“Loot boxes, the main paid virtual items, exist in most video games”) is dubious. Has a study found that the majority of video games contain non-free loot boxes? A later claim is that “‘free-to-play’ has become the dominant game profit model,” but the article cited – Tomic, 2017 – only makes that claim about the mobile gaming market, and is not a primary source for even that.

The manuscript attributes the RLI development to Brooks, but that work is authored by Brooks and Clark. Both authors should be credited in-text. Later text refers to them as Brooks et al., but the “et al.” should only be used when there are three or more authors.

Although the writing is readable, it has various slips throughout. For example, “publishers attempt to add various optional paid content” – there is no “attempt”; this is exactly what they do if they wish to do so. The manuscript later states that participants “will be rewarded,” which is the wrong tense. A general proofing by an outside reader may be helpful.

Experimental design

This study sought to provide a validated Chinese language version of the risky loot box index. The use of multiple samples is a strength, as are the protocols used for developing the translation. I found the goals of the study to be mostly fulfilled, though I am not confident that developing a translation of an instrument, in itself, is relevant and meaningful, or fills a knowledge gap. Ordinarily, I would expect this to be the first step in a larger study rather than a standalone contribution.

Very little information is presented about sampling. It’s clear both samples are convenience samples, but even so there could be more detail. For the cafe-based samplings, how many people declined to participate to give a sense for the participation rate? For the online recruitment, how were the links provided? There are different strategies that could be employed in soliciting participants through discussion forums.

It is reported that 148 invalid questionnaires were removed. The criteria for removal were provided, but no where does it explain which of the five criteria resulted in removal. How many cases were removed for each criteria?

The authors note in section 3.4 that gambling is illegal, so they “refer to Xiao’s suggestion.” This entire section seems vitally important and should be clarified with some additional information.

The authors state: “One month later, 50 people were randomly selected from the online sample to complete the scale again, and a test-retest reliability analysis was completed based on the results.” What was the response rate? How were they contacted? The description of the questionnaire referred to anonymity, so how could they be re-contacted?

Validity of the findings

Any concerns about the validity of the findings are contingent on more information regarding the concerns about the study design.

·

Basic reporting

This is a review of "Psychometric evaluation of the Chinese version of risky loot box index (RLI) and cross-sectional investigation among gamers of China" for PeerJ Life & Environment.

1. Basic Reporting

The language needs to be improved. For example, "the main paid virtual items," "Huge user groups," "not all games are able to sell virtual goods (can get profit from loot boxes)," and "it doesn't mean they are easier at risky loot 284 box usage" are not common English usage. I would recommend further proofreading and refinement.

"by Brooks" should be "Brooks and Clark"

"and supposed that loot boxes may be a trigger for problem gambling." No, that is only one possible explanation. It should not be stated on its own without considering alternatives: the other direction of the correlational relationship.

Line 78: The authors claim no research has been conducted on the RLI in China, which is not true: https://doi.org/10.1016/j.addbeh.2023.107860. I admit that is my work. That was published on 18 September 2023, and so the authors should have been able to find it if an adequate literature review was conducted. I understand the authors' study was undertaken in summer 2024.

Whilst I believe in the benefits of replications, the inaccurate claim to priority should be amended.

Given now that there are two independently translated version of the RLI in Chinese, I would ask the authors to identify translation differences and test which translation was better so that their study makes a unique contribution. The RLI we translated is publicly available at the data deposit link, as are the participant responses to the RLI.

Other loot box research has also been conducted before in China, and the authors should consider acknowledging them to provide a comprehensive literature review (those studies should have been considered when designing the current one). I note that the authors considered one of these studies for context and insights:
https://doi.org/10.1007/s10899-022-10148-0
https://doi.org/10.3389/fpsyt.2022.940281
https://doi.org/10.1017/bpp.2021.23
https://doi.org/10.1080/13600834.2022.2088062

That Chinese participants might not understand "loot boxes" is a point that was made in https://doi.org/10.1016/j.addbeh.2023.107860. It is nice to see the authors coming to the same conclusion. Did the authors consider the translation in that paper or was that paper not considered when designing the present research?

That gambling is strictly prohibited in Mainland China is not entirely correct because various lottery products are permitted. In addition, illegal gambling should also be acknowledged.

Experimental design

2. Experimental design

The amount of money paid to participants is very low, even for Chinese standards (1 CNY for 5 minutes of their time, so 12 CNY per hour). How does this potentially affect validity? e.g., people rushing to finish.

The questionnaire supposedly took 5 minutes for people to finish. However, there is a rather long consent form and also a lot of questions requiring a fair amount of reading, I would have expected it to take longer than 5 minutes. Were the authors surprised by the short length of time taken by participants? This further affects participants' pay.

Did the authors conduct power analysis? If not, why not?

Was the research preregistered? If not, why not?

I defer to other reviewers with relevant expertise to review the statistics and the instrument evaluation.

Validity of the findings

3. Validity of the findings

I defer to other reviewers with relevant expertise to review the statistics and the instrument evaluation.

Additional comments

4. General comments

I am satisfied with the data uploaded. I did not conduct the stats tests but did confirm it should be useable by English speakers.

However, I would ask the authors to provide the survey instruments, e.g., the translated RLI, in addition to the consent form.

Nothing further.

---

## Round 0.2 · Minor Revisions

We appreciate the effort you have put into addressing the reviewers' comments and improving the quality of your work.

While the manuscript has improved significantly, the reviewers have identified a few remaining points that need further clarification. We kindly ask you to address these points in a minor revision. Please find the reviewers' comments and our suggestions for improvement in the attached document.

We look forward to receiving your revised manuscript soon. Should you have any questions or need further clarification, please do not hesitate to contact us.

Thank you for your cooperation.

Reviewer 1 ·

Basic reporting

No comment.

Experimental design

No comment.

Validity of the findings

No comment.

Additional comments

I appreciate the revisions that the authors have made to the manuscript. However, I still have some reservations about the manuscript in its present state.

In my previous review, I questioned the validity of the statement that “Loot boxes, the main paid virtual items, exist in most video games.” The statement – revised based on another reviewer’s comment – now reads as “Nowadays, most video games use loot boxes as the main paid virtual items.” The authors have added some citations relevant to this claim later in the manuscript. However, the evidence still falls short of supporting this bold statement. The authors state in their response that “A total of 58.0% of the top games on the Google Play store contained loot boxes, 59.0% of the top iPhone games contained loot boxes and 36.0% of the top games on the Steam store contained loot boxes. Another study by Xiao et al. conducted a similar investigation in mainland China, they found that 91 of the 100 highest-grossing PRC iPhone games contain loot boxes.” These are all qualified statements, limiting the percentage to the “top games” on specific platforms, and one of them does not even reach the majority threshold. I still see this statement as unsupported. In the interest of finding common ground, I would suggest the reviewers consider replacing that sentence with the following or something similar: “Many of the top-selling video games include options to purchase loot boxes as paid virtual items.”

Other than that, however, I believe that the authors have made revisions or provided explanation to address the other concerns, and I thank them for their work.

·

Basic reporting

I am satisfied with the authors' detailed response. It is excellent to see more work outside of Western countries on loot boxes, which is an issue that affects people internationally.

Thanks also for your engagement with open science including preregistration and data sharing.

Personally, I am still unsure that 战利品箱 in Chinese or loot boxes in English is a sufficiently inclusive terminology. I think we can do better in the future. This could certainly be a direction for further research. Happy New Year!

Experimental design

No comment

Validity of the findings

No comment

Additional comments

No comment

---

## Round 0.3 · accepted · Accept

The authors have addressed all of the reviewers' comments. The revised version of the manuscript is suitable for publication.

Reviewer 1 ·

Basic reporting

No comment.

Experimental design

No comment.

Validity of the findings

No comment.

Additional comments

I appreciate the revisions that the authors have made to the manuscript. It is my assessment that the authors have made revisions or provided explanation to address the concerns raised, and I thank them for their work.